# Amazon-M2: A Multilingual Multi-locale Shopping Session Dataset for Recommendation and Text Generation

Wei Jin[12]*, Haitao Mao[3]*, Zheng Li[1], Haoming Jiang[1], Chen Luo[1],
Hongzhi Wen[3], Haoyu Han[3], Hanqing Lu[1], Zhengyang Wang[1], Ruirui Li[1],
Zhen Li[1], Monica Cheng[1], Rahul Goutam[1], Haiyang Zhang[1], Karthik Subbian[1],
Suhang Wang[4], Yizhou Sun[5], Jiliang Tang[3], Bing Yin[1], and Xianfeng Tang[1]

[1] Amazon.com     [2] Emory University     [3] Michigan State University
[4] The Pennsylvania State University     [5] University of California, Los Angeles

{joewjin,amzzhe,jhaoming,zhengywa,ruirul,luhanqin,cheluo}@amazon.com,
{amzzhn,chengxc,rgoutam,hhaiz,ksubbian,alexbyin,xianft}@amazon.com, wei.jin@emory.edu,
szw494@psu.edu,  yzsun@cs.ucla.edu, {haitaoma,wenhongz,hanhaoy1,tangjili}@msu.edu

## Abstract

Modeling customer shopping intentions is a crucial task for e-commerce, as it directly impacts user experience and engagement. Thus, accurately understanding customer preferences is essential for providing personalized recommendations. Session-based recommendation, which utilizes customer session data to predict their next interaction, has become increasingly popular. However, existing session datasets have limitations in terms of item attributes, user diversity, and dataset scale. As a result, they cannot comprehensively capture the spectrum of user behaviors and preferences. To bridge this gap, we present the *Amazon Multilingual Multi-locale Shopping Session Dataset*, namely *Amazon-M2*. It is the first multilingual dataset consisting of millions of user sessions from six different locales, where the major languages of products are English, German, Japanese, French, Italian, and Spanish. Remarkably, the dataset can help us enhance personalization and understanding of user preferences, which can benefit various existing tasks as well as enable new tasks. To test the potential of the dataset, we introduce three tasks in this work: (1) next-product recommendation, (2) next-product recommendation with domain shifts, and (3) next-product title generation. With the above tasks, we benchmark a range of algorithms on our proposed dataset, drawing new insights for further research and practice. In addition, based on the proposed dataset and tasks, we hosted a competition in the KDD CUP 2023[2] and have attracted thousands of users and submissions. The winning solutions and the associated workshop can be accessed at our website https://kddcup23.github.io/.

## 1   Introduction

In the era of information explosion, recommender systems have become a prevalent tool for understanding user preferences and reducing information overload [1, 2, 3, 4, 5, 6].  Traditionally, the majority of recommendation algorithms focus on understanding long-term user interests through utilizing user-profiles and behavioral records. However, they tend to overlook the user's current purpose

---

*Equal contribution.

[2]https://www.aicrowd.com/challenges/amazon-kdd-cup-23-multilingual-recommendation-challenge

37th Conference on Neural Information Processing Systems (NeurIPS 2023) Track on Datasets and Benchmarks.

which often has a dominant impact on user's next behavior. Besides, many recommendation algorithms require access to user profiles [7, 8, 9], which can be incomplete or even missing in real-world situations especially when users are browsing in an incognito mode. In these cases, only the most recent user interactions in the current session can be utilized for understanding their preferences. Consequently, the session-based recommendation has emerged as an effective solution for modeling user's short-term interest, focusing on a user's most recent interactions within the current session to predict the next product. Over the past few years, the session-based recommendation has gained significant attention and has prompted the development of numerous models [10, 11, 12, 13, 14, 15, 16, 17].

A critical ingredient for evaluating the efficacy of these methods is the session dataset. While numerous session datasets [18, 19, 20, 11, 21] have been carefully curated to meet the requirements of modeling user intent and are extensively employed for evaluating session-based recommender systems, they have several drawbacks. First, existing datasets only provide limited product attributes, resulting in incomplete product information and obscuring studies that leverage attribute information to advance the recommendation. Second, the user diversity within these datasets is limited and may not adequately represent the diversity of user-profiles and behaviors. Consequently, it can result in biased or less accurate recommendations, as the models may not capture the full range of customer preferences. Third, the dataset scale, particularly in terms of the product set, is limited, which falls short of reflecting real-world recommendation scenarios with vast product and user bases.

To break the aforementioned limitations, we introduce the *Amazon Multilingual Multi-Locale Shopping Session Dataset*, namely *Amazon-M2*, a large dataset of anonymized user sessions with their interacted products collected from multiple language sources at Amazon. Specifically, the dataset contains samples constructed from real user session data, where each sample contains a list of user-engaged products in chronological order. In addition, we provide a table of product attributes, which contains all the interacted products with their associated attributes such as title, brand, color, etc. Modeling such session data can help us better understand customers' shopping intentions, which is also the main focus of e-commerce. Particularly, the proposed dataset exhibits the following characteristics that make it unique from existing session datasets.

(a) **Rich semantic attributes**: *Amazon-M2* includes rich product attributes (categorical, textual, and numerical attributes) as product features including title, price, brand, description, etc. These attributes provide a great opportunity to accurately comprehend the user's interests. To our best knowledge, it is *the first session dataset to provide textual features*.

(b) **Large scale**: *Amazon-M2* is a large-scale dataset with millions of user sessions and products, while existing datasets only contain tens of thousands of products.

(c) **Multiple locales**: *Amazon-M2* collected data from diverse sources, i.e., six different locales including the United Kingdom, Japan, Italian, Spanish, French, and Germany. Thus, it provides a diverse range of user behaviors and preferences, which can facilitate the design of less biased and more accurate recommendation algorithms.

(d) **Multiple languages**: Given the included locales, *Amazon-M2* is special for its multilingual property. Particularly, six different languages (English, Japanese, Italian, Spanish, French, and German) are provided. It enables us to leverage recent advances such as language models [22, 23, 24] to model different languages in user sessions.

By utilizing this dataset, we can perform diverse downstream tasks for evaluating relevant algorithms in recommendation and text generation. Here, we focus on three different tasks, consisting of (1) next-product recommendation, (2) next-product recommendation with domain shifts, and (3) next-product title generation. **The first task** is the classic session-based recommendation which requires models to predict the ID of the next product, where the training dataset and test dataset are from the same domain. **The second task** is similar to the first task but requires the models to pre-train on the large dataset from large locales and transfer the knowledge to make predictions on downstream datasets from different domains (i.e., underrepresented locales). **The third task** is a novel task proposed by us, which asks models to predict the title of the next product which has never been shown in the training set. Based on these tasks, we benchmark representative baselines along with simple heuristic methods. Our empirical observations suggest that the representative baselines fail to outperform simple heuristic methods in certain evaluation metrics in these new settings. Therefore, we believe that *Amazon-M2* can inspire novel solutions for session-based recommendation and enable new opportunities for tasks that revolve around large language models and recommender systems.

Table 1: Comparison among popular session datasets. Note that "Diverse products" indicates whether the products are from diverse categories or only specific categories (e.g., fashion clothes).

| | #Train Sessions | #Test Sessions | #Products | Textual feat. | Multilingual | Multi-locale | Diverse products |
|---|---|---|---|---|---|---|---|
| Yoochoose [18] | 1,956,539 | 15,324 | 30,660 | ✗ | ✗ | ✗ | ✓ |
| Tmall [21] | 195,547 | 4,742 | 40,728 | ✗ | ✗ | ✗ | ✓ |
| Diginetica [25] | 186,670 | 15,963 | 43,097 | ✗ | ✗ | ✗ | ✓ |
| Dressipi [19] | 1,000,000 | 100,000 | 23,496 | ✗ | ✗ | ✓ | ✗ |
| *Amazon-M2* | 3,606,249 | 361,659 | 1,410,675 | ✓ | ✓ | ✓ | ✓ |

## 2 Dataset & Task Description

### 2.1 Dataset Description

Before we elaborate on the details of the proposed *Amazon-M2* dataset, we first introduce session-based recommendation. Given a user session, session-based recommendation aims to provide a recommendation on the product that the user might interact with at the next time step. As shown in Figure 1a, each session is represented as a chronological list of products that the user has interacted with. Specifically, we use $S = \{s_1, s_2, \ldots, s_n\}$ to denote the session dataset containing $n$ sessions, where each session is represented by $s = \{e_1, e_2, \ldots, e_t\}$ with $e_t$ indicating the product interacted by the user at time step $t$. In addition, let $V = \{v_1, v_2, \cdots, v_m\}$ denote a dictionary of unique products that appeared in the sessions, and each product is associated with some attributes.

Designed for session-based recommendation, *Amazon-M2* is a large-scale dataset composed of customer shopping sessions with interacted products. Specifically, the dataset consists of two components: (1) user sessions where each session is a list of product IDs interacted by the current user (Figure 1a), and (2) a table of products with each row representing the attributes of one product (Figure 1b). Particularly, the user sessions come from six different locales, i.e., the United Kingdom (UK), Japan (JP), German (DE), Spain (ES), Italian (IT), and France (FR). Given its multi-locale nature, the dataset is also multilingual: the textual attributes (e.g., title and description) of the products in the user sessions are in multiple languages, namely, English, Italian, French, Germany, and Spanish. Based on this dataset, we construct the training/test dataset for each task. A summary of our session dataset is given in Table 2. It includes the number of sessions, the number of interactions, the number of products, and the average session length for six different locales. We can find that UK, DE, and JP have approximately 10 times the number of sessions/products compared to ES, FR, and IT. More details about the collection process of the dataset can be found in Appendix B.

**Comparison with Existing Datasets.** We summarize the differences between existing session datasets (especially from session-based recommendation) and *Amazon-M2* in Table 1. First of all, *Amazon-M2* is the first dataset to provide textural information while other datasets majorly focus on the product ID information or categorical attributes. Without comprehensive product attributes, the recommendation models may struggle to capture the nuanced preferences of customers. Second, existing datasets only provide sessions from a single locale (or country) which limits their user diversity. Consequently, it may lead to biased or less accurate recommendations, as the models may not capture the full range of customer preferences. By contrast, our proposed *Amazon-M2* is collected from multiple locales and is multilingual in nature. Third, our proposed *Amazon-M2* provides a large number of user sessions and is on a much larger product scale, which can better reflect real-world recommendation scenarios with huge product bases.

### 2.2 Task Description

The primary goal of this dataset is to inspire new recommendation strategies and simultaneously identify interesting products that can be used to improve the customer shopping experience. We introduce the following three different tasks using our shopping session dataset.

(a) **Task 1. Next-product recommendation**. This task focuses on traditional session-based recommendation task, aiming to identify the next product of interest within a user session. Given a user session, the goal of this task is to predict the next product that the user will interact with. Note that the training/test data are from the same distribution of large locales (JP, UK, and DE).

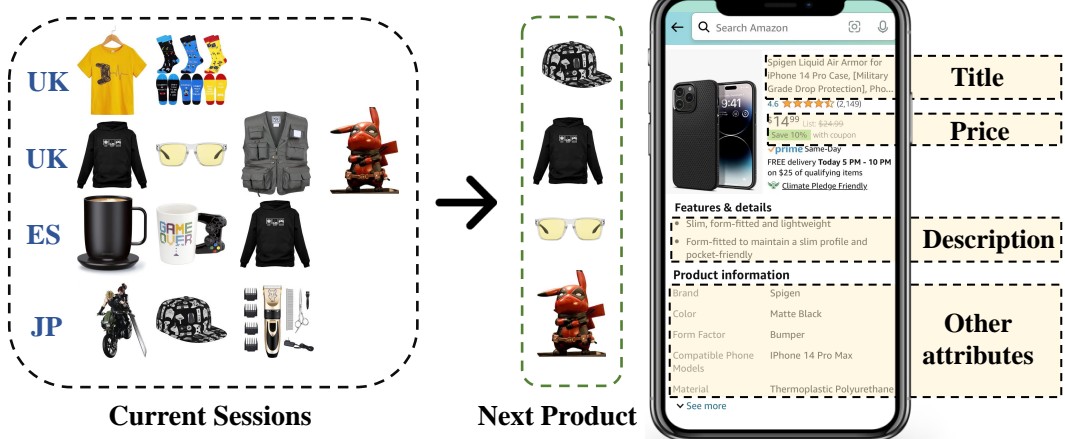

|                     |                    |
|---------------------|--------------------|
| **Current Sessions** | **Next Product**  |

(a) User sessions from different locales       (b) Product attributes

Figure 1: An illustration of the proposed *Amazon-M2* dataset. (a) A user session contains a list of previous products that the user has interacted with and the next product that the user is going to interact with. The user sessions come from multiple locales such as UK, ES, JP, etc. It is import to note that one product can appear in multiple locales. (b) The product attributes are publicly available and can be accessed on Amazon.com. Users can find information about a specific product by its ASIN number.

(b) **Task 2. Next-product recommendation with domain shifts**. This task is similar to Task 1 but advocates a novel setting of transfer-learning: practitioners are required to perform pretraining on a large pretraining dataset (user sessions from JP, UK, and DE) and then finetune and make predictions on the downstream datasets of underrepresented locales (user sessions from ES, IT, and FR). Thus, there is a domain shift between the pretraining dataset and the downstream dataset, which requires the model to transfer knowledge from large locales to facilitate the recommendation for underrepresented locales. This can address the challenge of data scarcity in these languages and improve the accuracy and relevance of the recommendations. More discussions on distribution shift can be found in Section 3.

(c) **Task 3. Next-product title generation**. This is a brand-new task designed for session datasets with textual attributes, and it aims to predict the title of the next product that the user will interact with within the current session. Notably, the task is challenging as the products in the test set do not appear in the training set. The generated titles can be used to improve cold-start recommendation and search functionality within the e-commerce platform. By accurately predicting the title of the next product, users can more easily find and discover items of interest, leading to improved user satisfaction and engagement.

The tasks described above span across the fields of recommender systems, transfer learning, and natural language processing. By providing a challenging dataset, this work can facilitate the development of these relevant areas and enable the evaluation of machine learning models in realistic scenarios.

## 3 Dataset Analysis

In this section, we offer a comprehensive analysis of the *Amazon-M2* dataset to uncover valuable insights. Our analysis covers several essential perspectives: long-tail phenomenon, product overlap between locales, session lengths, repeat pattern, and collaborative filtering pattern. Corresponding codes can be found here.

**Long-tail phenomenon** [26, 27] is a significant challenge in the session recommendation domain. It refers to the situation where only a handful of products enjoy high popularity, while the majority of products receive only a limited number of interactions. To investigate the presence of the long-tail phenomenon in *Amazon-M2* dataset, we analyze the distribution of product frequencies, as depicted

Table 2: Statistics of the multilingual shopping session dataset for training and test: the number of sessions, the number of products, the number of interactions, and the average session length.

| Language | #Products | Train | | | Test | | |
|---|---|---|---|---|---|---|---|
| | | #Sessions | #Interactions | Avg. Length | #Sessions | #Interactions | Avg. Length |
| UK | 500,180 | 1,182,181 | 4,872,506 | 4.1 | 115,936 | 466,265 | 4.0 |
| DE | 518,327 | 1,111,416 | 4,836,983 | 4.4 | 104,568 | 450,090 | 4.3 |
| JP | 395,009 | 979,119 | 4,388,790 | 4.5 | 96,467 | 434,777 | 4.5 |
| ES | 42,503 | 89,047 | 326,256 | 3.7 | 8,176 | 31,133 | 3.8 |
| FR | 44,577 | 117,561 | 416,797 | 3.5 | 12,520 | 48,143 | 3.9 |
| IT | 50,461 | 126,925 | 464,851 | 3.7 | 13,992 | 53,414 | 3.8 |
| Total | 1,410,675 | 3,606,249 | 15,306,183 | 4.2 | 361,659 | 1,483,822 | 4.2 |

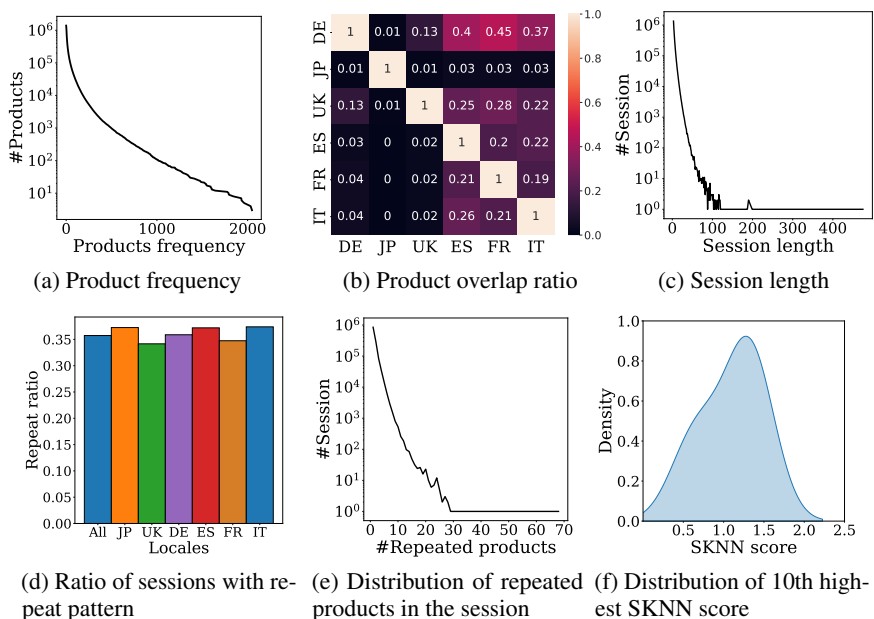

(a) Product frequency   (b) Product overlap ratio   (c) Session length

(d) Ratio of sessions with re-  (e) Distribution of repeated  (f) Distribution of 10th high-
peat pattern  products in the session  est SKNN score

Figure 2: Data analysis on *Amazon-M2*. (a)(c)(e) illustrate the long-tail phenomenon of product frequency, session length distribution, and the number of repeat products in a single session. (b) shows product overlap ratio between locales. (d) illustrates the proportion of sessions with repeat patterns in different locales. (f) shows most sessions can find relevant products of high SKNN scores.

in Figure 2a. The results clearly demonstrate the existence of a long-tail distribution, where the head of the distribution represents popular items and the tail consists of less popular ones. Furthermore, we observe that the long-tail phenomenon is also evident within each individual locale. For detailed experimental results regarding this phenomenon in each locale, please refer to Appendix B.2. The long-tail distribution makes it difficult to effectively recommend less popular products, as a small number of popular items dominate the recommendations.

**Product overlap ratio between locales** is the proportion of the same products shared by different locales. A large number of overlapping products indicates a better transferability potential when transferring the knowledge from one locale to the other. For example, cross-domain recommendation algorithms like [28] can then be successfully applied, which directly transfers the learned embedding of the overlapping products from popular locales to the underrepresented locales. We then examine product overlap between locales in *Amazon-M2* with the product overlap ratio. It is calculated as $\frac{|\mathcal{N}_a \cap \mathcal{N}_b|}{|N_a|}$, where $\mathcal{N}_a$ and $\mathcal{N}_b$ correspond to the products set of locale $a$ and $b$, respectively. In Figure 2b we use a heatmap to show the overlap ratio, where $x$ and $y$ axes stand for locale $a$ and $b$, respectively. From the figure, we make several observations: (1) For the products in the three large locales, i.e., UK, DE, and JP, there are not many overlapping products, except the one between UK and DE locales;

(2) Considering the product overlap ratio between large locales and underrepresented locales, i.e., ES, FR, and IT, we can see a large product overlapping, indicating products in the underrepresented domain also appear in the large locales. Particularly, the overlap ratio between small locales and DE can reach around 0.4. Thus, it has the potential to facilitate knowledge transfer from large locales and areas to underrepresented regions.

Notably, despite the existence of overlapping products between different locales, there still remains a large proportion of distinguished products in each locale, indicating the difficulty of transferability with distribution shift. Moreover, the multilingual property, where the product textual description from different locales is in different languages, also induces to distribution shift issue. Such a multilingual issue is a long-established topic in the NLP domain. For instance, [29, 30, 31] point out morphology disparity, tokenization differences, and negative transfer in the multilingual scenario, leading to distribution shift.

**Session length** is an important factor in the session recommendation domain. Typically, a longer session length may lead to the interest shift challenge problem [15] with difficulties in capturing multiple user interests in one single session. Most existing algorithms [13, 16] show a better performance on the shorter sessions while failing down on the longer ones. As shown in Figure 2c. We can observe that the session length also exhibits a long-tail distribution: most sessions are short while only few sessions are with a length larger than 100.

**Repeat pattern** [32, 33, 34, 35] is also an important user pattern, which refers to the phenomenon that a user repeatedly engages the same products multiple times in a single session. The presence of repeat patterns in recommender systems can potentially result in the system offering more familiar products that match users' existing preferences, which may lead to a less diverse and potentially less satisfying user experience. On the other hand, the repeat pattern is also an important property utilized in the graph-based session recommendation algorithms [13, 17, 36, 37]. Typically, those graph-based algorithms construct a session graph where each node represents a product and each edge indicates two products interacted by the user consecutively. Complicated session graphs with different structure patterns can be built when sessions exhibit evident repeat patterns. In Figure 2d, we report the proportion of sessions with repeat patterns for the six locales and we can observe that there are around 35% sessions with repeat patterns across different locales. Furthermore, we examine the number of repeat products in those sessions with repeat patterns and report results on the distribution of repeated products in Figure 2e. We make two observations: (1) the number of repeated products varies on different sessions; and (2) the number of repeated products in a session also follows the long-tail distribution where most sessions only appear with a few repeated products.

**Collaborative filtering pattern.** Collaborative filtering is a widely used technique that generates recommended products based on the behavior of other similar users. It is generally utilized as an important data argumentation technique to alleviate the data sparsity issue, especially for short sessions [38, 39]. Since *Amazon-M2* encompasses a much larger product set than existing datasets, we investigate whether collaborative filtering techniques can potentially operate in this challenging data environment. Specifically, we utilize the session collaborative filtering algorithm, Session-KNN (SKNN) [11], to identify sessions that are similar to the target user's current session. The similarity score of SKNN can be calculated in the following steps. First, for a particular session $s$, we first determines a set of its most similar sessions $\mathcal{N}(s) \subseteq S$ with the cosine similarity $\text{sim}(s, s_j) = |s \cap s_j| / \sqrt{|s||s_j|}$. Second, the score of a candidate product $e$ in similar sessions $\mathcal{N}(s)$ is then calculated by: $\text{score}(e, s) = \sum_{n \in \mathcal{N}(s)} \text{sim}(s, n) I_n(e)$, where the indicator function $I_n(e)$ is 1 if $n$ contains item $e$. Third, for each session $s$, we choose the 10th highest $\text{score}(e, s)$ to indicate the retrieval quality of candidate products. In Figure 2f, we show the distribution of the 10th highest SKNN scores for all sessions. A notable observation is that the majority of sessions exhibit a high SKNN score, hovering around 1. This finding suggests that for most sessions, it is possible to retrieve at least 10 similar products to augment the session data.

## 4 Benchmark on the Proposed Three Tasks

### 4.1 Task 1. Next-product Recommendation

In this task, we evaluate the following popular baseline models (deep learning models) in session-based recommendation on our proposed *Amazon-M2*, with the help of Recobole package [3]:

Table 3: Experimental results on Task 1, next product prediction.

| | MRR@100 | | | | Recall@100 | | | |
|---|---|---|---|---|---|---|---|---|
| | UK | DE | JP | Overall | UK | DE | JP | Overall |
| Popularity | 0.2302 | 0.2259 | 0.2766 | 0.2426 | 0.4047 | 0.4065 | 0.4683 | 0.4243 |
| GRU4Rec++ | 0.1656 | 0.1578 | 0.2080 | 0.1757 | 0.3665 | 0.3627 | 0.4185 | 0.3808 |
| NARM | 0.1801 | 0.1716 | 0.2255 | 0.1908 | 0.4021 | 0.4016 | 0.4559 | 0.4180 |
| STAMP | 0.2050 | 0.1955 | 0.2521 | 0.2159 | 0.3370 | 0.3320 | 0.3900 | 0.3512 |
| SRGNN | 0.1841 | 0.1767 | 0.2282 | 0.1948 | 0.3882 | 0.3862 | 0.4403 | 0.4032 |
| CORE | 0.1510 | 0.1510 | 0.1846 | 0.1609 | 0.5591 | 0.5525 | 0.5898 | 0.5591 |
| MGS | 0.1668 | 0.1739 | 0.2376 | 0.1907 | 0.5641 | 0.5479 | 0.4677 | 0.5194 |

Table 4: Experimental results on Task 2, next-product recommendation with domain shifts.

| | | MRR@100 | | | | Recall@100 | | | |
|---|---|---|---|---|---|---|---|---|---|
| | Methods | ES | FR | IT | Overall | ES | FR | IT | Overall |
| Heuristic | Popularity | 0.2854 | 0.2940 | 0.2706 | 0.2829 | 0.5118 | 0.5166 | 0.4914 | 0.5058 |
| Supervised training | GRU4Rec++ | 0.2538 | 0.2734 | 0.2420 | 0.2564 | 0.5817 | 0.5995 | 0.5753 | 0.5856 |
| | NARM | 0.2598 | 0.2805 | 0.2493 | 0.2632 | 0.5894 | 0.6061 | 0.5803 | 0.5920 |
| | STAMP | 0.2555 | 0.2752 | 0.2430 | 0.2578 | 0.5162 | 0.5365 | 0.5116 | 0.5217 |
| | SRGNN | 0.2627 | 0.2797 | 0.2500 | 0.2640 | 0.5426 | 0.5543 | 0.5322 | 0.5429 |
| | CORE | 0.1978 | 0.2204 | 0.1941 | 0.2045 | 0.6931 | 0.7215 | 0.6937 | 0.7034 |
| | MGS | 0.2491 | 0.2775 | 0.2411 | 0.2560 | 0.5829 | 0.5811 | 0.5689 | 0.5766 |
| Pretraining & finetuning | GRU4Rec++ | 0.2601 | 0.2856 | 0.2480 | 0.2646 | 0.5989 | 0.6236 | 0.5898 | 0.6042 |
| | NARM | 0.2733 | 0.2917 | 0.2564 | 0.2734 | 0.6094 | 0.6315 | 0.5974 | 0.6127 |
| | STAMP | 0.2701 | 0.2878 | 0.2584 | 0.2719 | 0.4792 | 0.4988 | 0.4637 | 0.4803 |
| | SRGNN | 0.2747 | 0.2980 | 0.2612 | 0.2779 | 0.5835 | 0.6101 | 0.5714 | 0.5883 |
| | CORE | 0.1685 | 0.1839 | 0.1632 | 0.1720 | 0.6946 | 0.7156 | 0.6851 | 0.6985 |
| | MGS | 0.2612 | 0.2870 | 0.2693 | 0.2722 | 0.5747 | 0.6133 | 0.5812 | 0.5913 |

- GRU4REC++ [40] is an improved model based on GRU4Rec which adopts two techniques to improve the performance of GRU4Rec, including a data augmentation process and a method to account for shifts in the 1input data distribution
- NARM [15] employs RNNs with attention mechanism to capture the user's main purpose and sequential behavior.
- STAMP [16] captures user's current interest and general interests based on the last-click product and whole session, respectively.
- SRGNN [13] is the first to employ GNN layer to capture user interest in the current session.
- CORE [41] ensures that sessions and items are in the same representation space via encoding the session embedding as a linear combination of item embeddings.
- MGS [42] incorporates product attribute information to construct a mirror graph, aiming to learn better preference understanding via combining session graph and mirror graph. Notably, MGS can only adapt categorical attributes. Therefore, we discretize the price attribute as the input feature.

In addition, we include a simple yet effective method, Popularity, by simply recommending all users the most popular products. We utilize Mean Reciprocal Rank@K (MRR@100) and Recall@100 to evaluate various recommendation algorithms. More results on NDCG@100 metric can be found in Appendix C. Corresponding codes can be found here.

**Results & Observations.** The experiment results across different locales can be found in Table 3. We can observe that the popularity heuristic generally outperforms all the deep models with respect to both MRR and Recall. The only exception is that CORE achieves better performance on Recall. On one hand, the success of the popularity heuristic indicates that the product popularity is a strong bias for this dataset. On the other hand, it indicates that the large product set in *Amazon-M2* poses great challenges in developing effective recommendation algorithms. Thus, more advanced recommendation strategies are needed for handling the challenging *Amazon-M2* dataset.

## 4.2 Task 2. Next-product Recommendation with Domain Shifts

The purpose of Task 2 is to provide next-product recommendations for underrepresented locales, specifically ES, IT, and FR. To evaluate the baseline methods, we consider two training paradigms. (1) Supervised training, which involves directly training models on the training data on ES, IT, and FR and then testing them on their test data. The goal of this paradigm is to evaluate the models' efficacy when no other data is available. (2) Pretraining & finetuning, which begins with pretraining the model using data from large locales, i.e., JP, UK, and DE, and then finetuning this pretrained model on the training data from ES, IT, and FR. This approach tests the models' ability to transfer knowledge from one source to another. In both paradigms, we incorporate the same baseline models used in Task 1, which include GRU4REC++ NARM, STAMP, SRGNN, and CORE. Additionally, we compare these methods with the popularity heuristic.

**Results & Observations.** The results across different locales can be found in Table 4. From the results, we have the following observations: (1) In the context of supervised training, the popularity heuristic outperforms all baselines in terms of MRR on underrepresented locales, which is consistent with observations from Task 1. Interestingly, despite this, most deep learning models surpass the popularity heuristic when considering Recall. This suggests a promising potential for deep learning models: they are more likely to provide the correct recommendations among the retrieved candidate set while the rankings are not high. (2) Finetuning most methods tends to enhance the performance in terms of both MRR and Recall compared to supervised training alone. This demonstrates that the knowledge of large locales can be transferred to underrepresented locales despite the existence of domain shifts. Importantly, it highlights the potential of adopting pretraining to enhance the performance on locales with limited data in the *Amazon-M2* dataset.

**Ablation Study.** Notably, the aforementioned methods utilized random initialization as the product embeddings. To enhance the model's ability to transfer knowledge across different locales, we can initialize product embeddings with embeddings from textual attributes. Specifically, we introduce SRGNNF and GRU4RecF to enable SRGNN and GRU4Rec++ to leverage the textual attributes (i.e., product title), respectively. To obtain the text embedding, we opt for pre-trained Multilingual Sentence BERT [43] to generate additional item embeddings from the item titles. As shown in Table 5, this straightforward approach has unintended effects in our ablation study. This could be due to the fact that the pre-training data for Multilingual Sentence BERT does not align with the text of product titles. Therefore, how to effectively utilize textual features remains an open question. We look forward to more research concerning the integration of language models with session recommendation.

## 4.3 Task 3. Next-product Title Generation

In this task, the goal is to generate the title of the next product of interest for the user, which is a text generation task. Therefore, we adopt the bilingual evaluation understudy (BLEU) score, a classic metric in natural language generation, as the evaluation metric. For the same reason, in this task, we explore the effectiveness of directly applying language models as a baseline recommendation model. We fine-tune a well-known multilingual pre-trained language model, mT5 [44], using a generative objective defined on our dataset. Specifically, the language model receives the titles of the most recent $K$ products and is trained to predict the title of the next product. We compare the performance by varying $K$ from 1 to 3, while also investigating the impact of parameter size. Additionally, we offer a simple yet effective baseline approach, namely Last Product Title, in which the title for the next product is predicted as that of the last product. We randomly select $10\%$ of training data for validation, and report the BLEU scores of different methods on validation and test set in Table 6.

Table 5: Ablation study on whether models can effectively leverage textual features.

| | MRR@100 | | | | Recall@100 | | | |
|---|---|---|---|---|---|---|---|---|
| | ES | FR | IT | Overall | ES | FR | IT | Overall |
| SRGNN | 0.2627 | 0.2797 | 0.2500 | 0.2640 | 0.5426 | 0.5543 | 0.5322 | 0.5429 |
| SRGNNF | 0.2107 | 0.2546 | 0.2239 | 0.2303 | 0.4541 | 0.5132 | 0.4788 | 0.4976 |
| GRU4Rec++ | 0.2538 | 0.2734 | 0.2420 | 0.2564 | 0.5817 | 0.5995 | 0.5753 | 0.5856 |
| GRU4RecF | 0.2303 | 0.2651 | 0.2303 | 0.2427 | 0.4976 | 0.5631 | 0.5353 | 0.5454 |

| T5 | Ground Truth |
|---|---|
| MuyDoux Funda para Xiaomi Mi Pad 5 / 5 Pro 11 Pulgadas 2021, Tapa Frontal Lisa y Reverso Suave, Encendido/Apagado automático, Funda Ligera y Delgada de Tres Pliegues. Color: Negro | MuyDoux Funda para Xiaomi Mi Pad 5 / 5 Pro 11 Pulgadas 2021, Tapa Frontal Lisa y Cubierta Trasera Suave, Auto Sueño/Estela, Carcasa Ligera y Delgada para Mi Xiaomi Pad 5 / 5 Pro 5G, Oro Rosa |
| Amazon Brand - Umi Colchón de Microfibra,Cubrecolchón,Antialérgico,Suave -(135x190cm) | Amazon Brand - Umi Colchón de Microfibra,Cubrecolchón,Antialérgico,Suave -(180x200cm) |
| Raid Spray Insecticida - Aerosol para Moscas y Mosquitos, Eucalipto. Eficacia Inmediata, Incoloro, 400 ml | Raid ® Spray Insecticida - Aerosol para moscas y mosquitos, Frescor Natural. Eficacia inmediata. Pack de 3 Unidades, 600ml |
| Rapesco 1498 Carpeta Sobre Portafolios Plástica con Broche de Presión, Tamaño A5, Colores Surtidos, Paquete de 25 | Rapesco 1494 Carpeta Sobre Portafolios A4+ horizontal, Colores Surtidos Translucídos, 20 unidades |

Figure 3: Comparison between ground truth title and product titles generated by mT5-small, K=1.

**Results & Observations.** The results in Table 6 demonstrate that for the mT5 model, extending the session history length ($K$) does not contribute to a performance boost. Furthermore, the simple heuristic, Last Product Title, surpasses the performance of the finetuned language model. This observation highlights two issues: (1) The last product bears significant importance, as the user's next interest is often highly correlated with the most recent product. This correlation becomes

Table 6: BLEU scores in Task 3.

|  | Validation | Test |
|---|---|---|
| mT5-small, $K = 1$ | 0.2499 | 0.2265 |
| mT5-small, $K = 2$ | 0.2401 | 0.2176 |
| mT5-small, $K = 3$ | 0.2366 | 0.2142 |
| mT5-base, $K = 1$ | 0.2477 | 0.2251 |
| Last Product Title | 0.2500 | 0.2677 |

particularly evident when the product title serves as the prediction target. For instance, the next product title can have a large overlap with the previous product (e.g., same product type or brand). (2) The mT5 model did not function as effectively as expected, potentially due to a mismatch between the pre-training and downstream texts. Thus, even an escalation in the model's size (from mT5-small to mT5-base) did not result in a performance gain. This suggests the necessity for a domain-specific language model to leverage text information effectively. Moreover, we illustrate some examples of generated results compared with the ground truth title in Figure 3 to intuitively indicate the quality of the generated examples. We note that the generated titles look generally good, nonetheless, the generated ones still lack details, especially numbers,e.g., (180x200cm) in example 2. In addition, we anticipate that larger language models such as GPT-3/GPT-4 [45, 46] could achieve superior performance in this task, given their exposure to a more diverse corpus during pre-training. We reserve these explorations for future work.

## 5 Discussion

In this section, we discuss the new opportunities that the proposed *Amazon-M2* dataset can introduce to various research topics in academic study and industrial applications. Due to the space limit, we only list five topics in this section while leaving others in Appendix B.4.

**Pre-Training & Transfer Learning.** Our dataset comprises session data from six locales with different languages, three of which have more data than the remaining three. This property presents a unique opportunity to investigate pre-training and transfer learning techniques for recommendation algorithms [14, 47, 48], as demonstrated in Task 2. Due to the inherent sparsity of session data [49], building accurate models that can capture users' preferences and interests based on their limited interactions can be challenging, particularly when the number of sessions is small. One solution to address the data sparsity problem is to transfer knowledge from other domains. Our dataset is advantageous in this regard because it provides data from sources (locales/languages), which enables the transfer of user interactions from other locales, potentially alleviating the data sparsity issue.

**Large Language Models in Recommendation.** The rich textual features in *Amazon-M2* give researchers a chance for leveraging large language models (LLMs) [50, 46, 45, 51] in providing recommendations. For example, there has been a growing emphasis on evaluating the capability of ChatGPT [45] in the context of recommender systems [52, 53, 54, 55, 56, 57, 58, 59, 60]. The proposed *Amazon-M2* presents an excellent opportunity for researchers to explore and experiment with their ideas involving LLMs and recommendation.

**Text-to-Text Generation.** Our dataset presents ample opportunities for exploring natural language processing tasks, including text-to-text generation. In Task 3, we introduced a novel task that involves predicting the title of the next product, which requires the recommender system to generate textual results, rather than simply predicting a single product ID. This new task resembles to *Question Answering* and allows us to leverage advanced text-to-text generation techniques [61], such as the popular GPT models [62, 51], which have demonstrated significant success in this area.

**Cross-Lingual Entity Alignment.** Entity alignment is a task that aims to find equivalent entities across different sources [63, 64, 65]. Our dataset provides a good resource for evaluating various entity alignment algorithms, as it contains entities (products) from different locales and languages. Specifically, the dataset can be used to test the performance of entity alignment algorithms in cross-lingual settings, where the entities to be aligned are expressed in different languages.

**Graph Neural Networks.** As powerful learning tools for graph data, graph neural networks (GNNs) [66, 67, 68] have tremendously facilitated a wide range of graph-related applications including recommendation [13, 4], computation biology [69, 70], and drug discovery [71]. Our dataset provides rich user-item interaction data which can be naturally represented as graph-structured data. Thus, it is a good tool for evaluating various GNNs and rethinking their development in the scenarios of recommendation and transfer learning. In addition, the abundant textual and structural data in *Amazon-M2* provides a valuable testing platform for the recent surge of evaluating LLMs for graph-related tasks [72, 73, 74, 75, 76].

**Item Cold-Start Problem.** The item cold-start problem [77, 38] is a well-known challenge in recommender systems, arising when a new item is introduced into the system, and there is insufficient data available to provide accurate recommendations. However, our dataset provides rich items attributes including detailed textual descriptions, which offers the potential to obtain excellent semantic embeddings for newly added items, even in the absence of user interactions. This allows for the development of a more effective recommender system that places greater emphasis on the semantic information of the items, rather than solely relying on the user's past interactions. Therefore, by leveraging this dataset, we can overcome the cold-start problem and deliver better diverse recommendations, enhancing the user experience.

**Data Imputation.** Research on deep learning requires large amounts of complete data, but obtaining such data is almost impossible in the real world due to various reasons such as damages to devices, data collection failures, and lost records. Data imputation [78] is a technique used to fill in missing values in the data, which is crucial for data analysis and model development. Our dataset provides ample opportunities for data imputation, as it contains entities with various attributes. By exploring different imputation methods and evaluating their performance on our dataset, we can identify the most effective approach for our specific needs.

# 6 Conclusion

This paper presents the *Amazon-M2* dataset, which aims to facilitate the development of recommendation strategies that can capture language and location differences. *Amazon-M2* provides rich semantic attributes including textual features, encompasses a large number of sessions and products, and covers multiple locales and languages. These qualities grant machine learning researchers and practitioners considerable flexibility to explore the data comprehensively and evaluate their models thoroughly. In this paper, we introduce the details of the dataset and provide a systematic analysis of its properties. Furthermore, through empirical studies, we highlight the limitations of existing session-based recommendation algorithms and emphasize the immense potential for developing new algorithms with *Amazon-M2*. We firmly believe that this novel dataset provides unique contributions to the pertinent fields of recommender systems, transfer learning, and large language models. The detailed discussion on broader impact and limitation can be found in Appendix D.

# 7 Acknowledgement

We want to thank AWS and amazon science for their support. We also appreciate the help from Yupeng Hou, in adding our dataset into RecBole.

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

# A  Experimental Setup

**Data Splits.** Following the setting in our KDDCUP competition[3], the *Amazon-M2* dataset contains three splits: training, phase-1 test and phase-2 test. For the purpose of model training and selection, we further split the original training set into 90% sessions for development (used for training), and 10% sessions for validation. Note that the numbers in Tables 3, 4, and 5 of the main content are for validation performance. Without specific mention, the test set mentioned in the main content indicates the phase-1 test. Due to the page limitation of main content, we defer the performances on the phase-2 test set to the appendix.

**Hyperparameter Settings.** The hyper-parameters of all the models are tuned based on the performance of the validation set. For Task 1 and Task 2, we follow the suggested hyper-parameter range to search for the optimal settings provided by Recbole toolkits [79]. By default, we only use the product ID to train the models since most of the popular session-based recommendation baselines are ID-based methods. We leave the exploration of other rich attributes such as price, brand, and description as future work. Specifically, the search ranges for different models are outlined below:

- GRU4REC++ :  learning_rate [0.01,0.001,0.0001], dropout_prob:  [0.0,0.1,0.2,0.3,0.4,0.5], num_layers: [1,2,3], hidden_size: [128].
- NARM: learning_rate: [0.01,0.001,0.0001], hidden_size: [128], n_layers: [1,2], dropout_probs: [’[0.25,0.5]’,’[0.2,0.2]’,’[0.1,0.2]’].
- STAMP: learning_rate: [0.01,0.001,0.0001]
- SRGNN: learning_rate: [0.01,0.001,0.0001], step: [1, 2].
- CORE: learning_rate:  [0.001, 0.0001], n_layers:  [1, 2], hidden_dropout_prob:  [0.2, 0.5], attn_dropout_prob: [0.2, 0.5].
- GRU4RECF: learning_rate: [0.01,0.001,0.0001], num_layers: [1, 2].
- SRGNNF: learning_rate: [0.01,0.001,0.0001], step: [1, 2].
- MGS: learning_rate: [0.01,0.001,0.0001]

For Task 3, we adopted the code example provided by HuggingFace[4] which supports text-to-text generation. Moreover, we tune the following hyperparameters in mT5: weight_decay in the range of {0, 1e-8}, learning_rate in the range of {2e-5, 2e-4}, num_beams in the range of {1, 5}. Additionally, we set the training batch size to 12, and the number of training epochs to 10.

**Hardware and Software Configurations.** We perform experiments on one server with 8 NVIDIA RTX A6000 (48 GB) and 128 AMD EPYC 7513 32-Core Processor @ 3.4 GHZ. The operating system is Ubuntu 20.04.1.

**Metric details:**   **Mean Reciprocal Rank(MRR)@K** is a metric used in information retrieval and recommendation systems to measure the effectiveness of a model in providing relevant results. MRR is computed with the following two steps: (1) calculate the reciprocal rank. The reciprocal rank is the inverse of the position at which the first relevant item appears in the list of recommendations. If no relevant item is found in the list, the reciprocal rank is considered 0. (2) average of the reciprocal ranks of the first relevant item for each session.

$$\text{MRR}@K = \frac{1}{N} \sum_{t \in T} \frac{1}{\text{Rank}(t)}, \tag{1}$$

where $\text{Rank}(t)$ is the rank of the ground truth on the top $K$ result ranking list of test session t, and if there is no ground truth on the top K ranking list, then we would set $\frac{1}{\text{Rank}(t)} = 0$. MRR values range from 0 to 1, with higher values indicating better performance. A perfect MRR score of 1 means that the model always places the first relevant item at the top of the recommendation list. An MRR score of 0 implies that no relevant items were found in the list of recommendations for any of the queries or users.

---

[3] https://www.aicrowd.com/challenges/amazon-kdd-cup-23-multilingual-recommendation-challenge
[4] https://github.com/huggingface/transformers/blob/main/examples/pytorch/summarization/run_summarization_no_trainer.py

**Hit@K** is the proportion of the correct test product within a top k position in the recommended ranking list, defined as:

$$\text{Hit@K} = \frac{n_{\text{hit}}}{N} \tag{2}$$

where $N$ denotes the number of test sessions. $n_{\text{hit}}$ is the number of test sessions with the next product in the top $K$ of the ranked list.

**Normalized Discounted Cumulative Gain(NDCG@K)** quantifies the effectiveness of ranked lists by considering both the relevance of items and their positions within the list up to a specified depth $K$. This metric facilitates fair comparisons of ranking algorithms across different datasets and scenarios, as it normalizes the Discounted Cumulative Gain (DCG) score to a standardized range, typically between 0 and 1.

## B  More Dataset Details

### B.1  Dataset Collection

The *Amazon-M2* dataset is a collection of anonymous user session data and product data from the Amazon platform. Each session represents a list of products that a user interacted with during a 30-minute active window. Note that the product list in each session is arranged in chronological order, with each product represented by its ASIN number. Users can search for Amazon products using their ASIN numbers[5] and obtain the corresponding product attributes. We include the following product attributes:

- ASIN (id): ASIN stands for Amazon Standard Identification Number. It is a unique identifier assigned to each product listed on Amazon's marketplace, allowing for easy identification and tracking.
- Locale: Locale refers to the specific geographical or regional settings and preferences that determine how information is presented to users on Amazon's platform.
- Title: The Title attribute represents the name or title given to a product, book, or creative work. It provides a concise and identifiable name that customers can use to search for or refer to the item.
- Brand: The Brand attribute represents the manufacturer or company that produces the product. It provides information about the brand reputation and can influence a customer's purchasing decision based on brand loyalty or recognition.
- Size: Size indicates the dimensions or physical size of the product. It is useful for customers who need to ensure that the item will fit their specific requirements or space constraints.
- Model: The Model attribute refers to a specific model or version of a product. It helps differentiate between different variations or versions of the same product from the same brand.
- Material Type: Material Type indicates the composition or main material used in the construction of the product. It provides information about the product's primary material, such as metal, plastic, wood, etc.
- Color Text: Color Text describes the color or color variation of the product. It provides information about the product's appearance and helps customers choose items that match their color preferences.
- Author: Author refers to the individual or individuals who have written a book or authored written content. It helps customers identify the creator of the work and plays a significant role in book purchasing decisions.
- Bullet Description (desc): Bullet Description is a concise and brief description of the product's key features, benefits, or selling points. It highlights the most important information about the item in a clear and easily scannable format.

The dataset spans a period of 3 weeks, with the first 2 weeks designated as the training set and the remaining week as the test set. To enhance evaluation, the test set is further randomly divided into two equal subsets, i.e., phase-1 test and phase-2 test.

### B.2  Additional data analysis

The additional analysis of the session length and the repeat pattern on each individual locale can be found in Figure 4 and 5, respectively. We can still observe evident long-tail distributions for both

---

[5]For instance, if the product is available in the US, users can access the product by using the following link: https://www.amazon.com/dp/ASIN_Number

product frequency and repeat pattern across the six locales, which is consistent with the observation that we made on the whole dataset in Section 3.

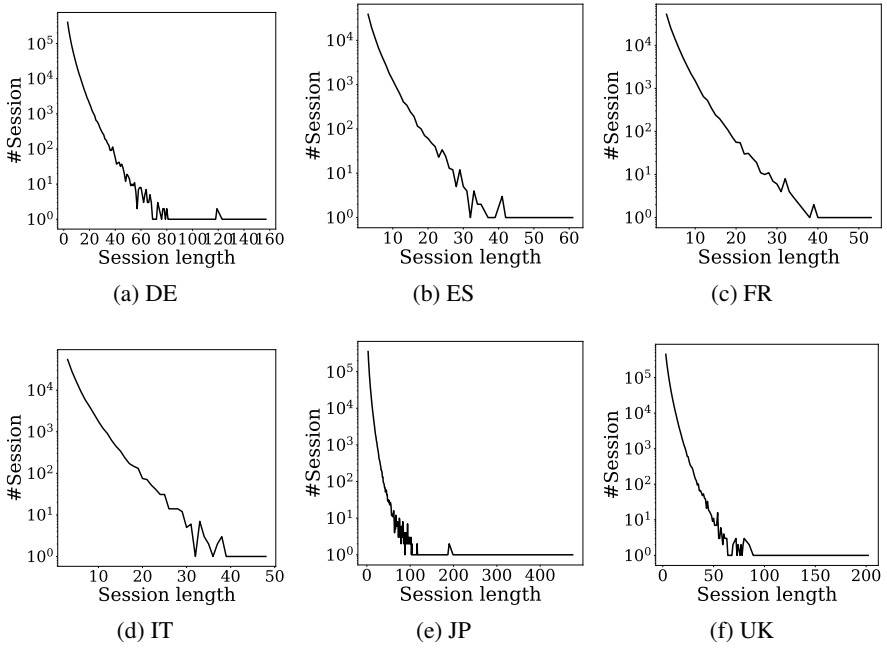

Figure 4: Session length w.r.t. locales where the x-axis corresponds to the session length (the number of items in a session), the y-axis indicates the number of sessions with the corresponding session length. A clear long-tail phenomenon can be found, where only a few sessions show a session length of more than 100.

## B.3 License

The *Amazon-M2* dataset can be freely downloaded at `https://www.aicrowd.com/challenges/amazon-kdd-cup-23-multilingual-recommendation-challenge/problems/task-1-next-product-recommendation/dataset_files` and used under the license of Apache 2.0. The authors agree to bear all responsibility in case of violation of rights, etc.

## B.4 Extended Discussion

**Item Cold-Start Problem.** The item cold-start problem [77, 38] is a well-known challenge in recommender systems, arising when a new item is introduced into the system, and there is insufficient data available to provide accurate recommendations. However, our dataset provides rich items attributes including detailed textual descriptions, which offers the potential to obtain excellent semantic embeddings for newly added items, even in the absence of user interactions. This allows for the development of a more effective recommender system that places greater emphasis on the semantic information of the items, rather than solely relying on the user's past interactions. Therefore, by leveraging this dataset, we can overcome the cold-start problem and deliver better diverse recommendations, enhancing the user experience.

**Data Imputation.** Research on deep learning requires large amounts of complete data, but obtaining such data is almost impossible in the real world due to various reasons such as damages to devices, data collection failures, and lost records. Data imputation [78] is a technique used to fill in missing values in the data, which is crucial for data analysis and model development. Our dataset provides ample opportunities for data imputation, as it contains entities with various attributes. By exploring different imputation methods and evaluating their performance on our dataset, we can identify the most effective approach for our specific needs.

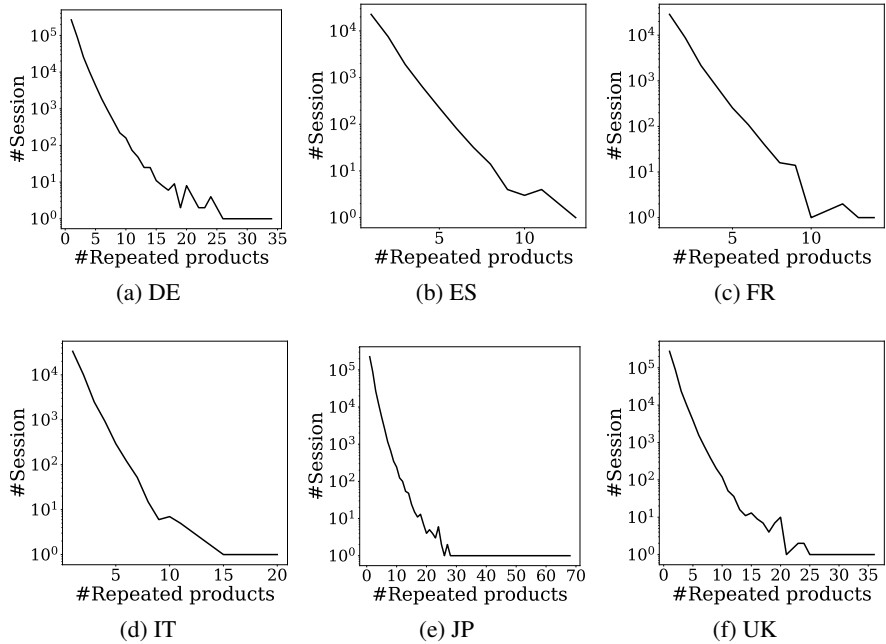

Figure 5: The number of repeat items w.r.t. locales where the x-axis corresponds to the number of repeat items, the y-axis indicates to the number of session with the corresponding number of repeat items. Notably, we exclude those sessions with no repeat patterns. A clear long-tail phenomenon can be found, where only a few sessions show many repeat items.

## C  More Experimental Results

### C.1  Additional results on NDCG@100

We provide additional NDCG@100 results on task 1 and 2. Results are shown in Table 7 and **??**, respectively. A similar observation can be found with Recall@100 metric.

### C.2  Task 1. Next-product Recommendation

In this subsection, we provide the model performance comparison on the phase-1 test and phase-2 test in Table 8 and Table 9, respectively. We can have similar observations as we made in Section 4.1: the popularity heuristic generally outperforms the deep models with respect to both MRR and Recall, with the only exception that CORE achieves better performance on Recall. This suggests that the popularity heuristic is a strong baseline and the challenging *Amazon-M2* dataset requires new recommendation strategies to handle. We believe that it is potentially helpful to design strategies that can effectively utilize the available product attributes.

### C.3  Task 2. Next-product Recommendation with Domain Shifts

We report the mode performances on the phase-1 test and phase-2 test in Table 10 and Table 11, respectively. Note that we omit the supervised training results since we have already identified that finetuning can significantly improve it. From the tables, we arrive at a similar observation as presented in Section 4.2 that the finetuned deep models generally outperform the popularity heuristic in Recall but underperform it in MRR. This illustrates that the deep models have the capability to retrieve a substantial number of pertinent products, but they fall short in appropriately ranking them. As a result, there is a need to enhance these deep models further in order to optimize their ranking efficacy.

Table 7: NDCG@100 results on Task 1.

|  | UK | DE | JP | Overall |
|---|---|---|---|---|
| Popularity | 0.2065 | 0.1993 | 0.2515 | 0.2175 |
| GRU4Rec++ | 0.2255 | 0.2184 | 0.2733 | 0.2374 |
| NARM | 0.2336 | 0.2248 | 0.2823 | 0.2452 |
| STAMP | 0.2262 | 0.2198 | 0.2726 | 0.2379 |
| SRGNN | 0.2325 | 0.2345 | 0.2715 | 0.2448 |
| CORE | 0.2698 | 0.2667 | 0.3204 | 0.2839 |
| MGS | 0.2314 | 0.2357 | 0.2782 | 0.2471 |

Table 8: Experimental results on Task 1 phase-1 test set.

|  | MRR@100 | | | | Recall@100 | | | |
|---|---|---|---|---|---|---|---|---|
|  | UK | DE | JP | Overall | UK | DE | JP | Overall |
| Popularity | 0.2723 | 0.2746 | 0.3196 | 0.2875 | 0.4940 | 0.5261 | 0.5652 | 0.5262 |
| GRU4Rec++ | 0.2094 | 0.2082 | 0.2527 | 0.2222 | 0.4856 | 0.5192 | 0.5416 | 0.5137 |
| NARM | 0.2235 | 0.2233 | 0.2705 | 0.2378 | 0.5220 | 0.5594 | 0.5814 | 0.5524 |
| STAMP | 0.2398 | 0.2398 | 0.2888 | 0.2547 | 0.4265 | 0.4538 | 0.4867 | 0.4538 |
| SRGNN | 0.2240 | 0.2211 | 0.2670 | 0.2361 | 0.4986 | 0.5311 | 0.5540 | 0.5262 |
| CORE | 0.1777 | 0.1797 | 0.2103 | 0.1882 | 0.6513 | 0.6927 | 0.7009 | 0.6801 |

## C.4 Task 3. Next-product Title Prediction

We expand Table 6 to include the results on phase-2 test and the full results are shown in Table 12. From the table, we have the same observations as we made in Section 4.3: (1) Extending the session history length ($K$) does not contribute to a performance boost, and (2) The simple heuristic of Last Product Title outperforms all other baselines. It calls for tailored designs of language models for this challenging task.

## D  Limitation & Broader Impact

The release of the *Amazon-M2* dataset brings several potential broader impacts and research opportunities in the field of session-based recommendation and language modeling. It provides the potential for research in the session recommendation domain to access the rich semantic attributes and knowledge from multiple locales, enabling better recommendation systems for diverse user populations.

While the *Amazon-M2* dataset offers significant research potential, it is crucial to consider the certain limitations associated with its use. Despite efforts that have been made to include diverse user behaviors and preferences with multiple locales and languages, it may not capture the full linguistic and cultural diversity of all regions. Moreover, the dataset can be only collected within the Amazon platform, which may not fully capture the diversity of user behaviors in other domains or platforms, leading to a potentially biased conclusion and may not hold true in different contexts.

We also carefully consider the broader impact from various perspectives such as fairness, security, and harm to people. There may be one potential fairness issue, Amazon-2M may inadvertently reinforce biases if specific demographics are disproportionately represented or underrepresented in different regions. Such issue may happen as we promote diversity with sessions from more locales. While there might be fairness concerns, we assert that the Amazon-M2 dataset underscores the need for progress in fair recommendation development and serves as an effective platform for assessing fair recommendation approaches.

Table 9: Experimental results on Task 1 phase-2 test set.

| | MRR@100 | | | | Recall@100 | | | |
|---|---|---|---|---|---|---|---|---|
| | UK | DE | JP | Overall | UK | DE | JP | Overall |
| Popularity | 0.2711 | 0.2754 | 0.3205 | 0.2875 | 0.4937 | 0.5283 | 0.5660 | 0.5271 |
| GRU4Rec++ | 0.2081 | 0.2097 | 0.2533 | 0.2224 | 0.4843 | 0.5220 | 0.5420 | 0.5143 |
| NARM | 0.2219 | 0.2235 | 0.2720 | 0.2377 | 0.5209 | 0.5624 | 0.5786 | 0.5522 |
| STAMP | 0.2387 | 0.2402 | 0.2894 | 0.2546 | 0.4234 | 0.4585 | 0.4864 | 0.4541 |
| SRGNN | 0.2224 | 0.2224 | 0.2695 | 0.2367 | 0.4974 | 0.5336 | 0.5529 | 0.5262 |
| CORE | 0.1755 | 0.1807 | 0.2111 | 0.1880 | 0.6518 | 0.6966 | 0.7002 | 0.6813 |

Table 10: Experimental results on Task 2 phase-1 test.

| | | MRR@100 | | | | Recall@100 | | | |
|---|---|---|---|---|---|---|---|---|---|
| | Methods | ES | FR | IT | Overall | ES | FR | IT | Overall |
| Heuristic | Popularity | 0.2934 | 0.2968 | 0.2887 | 0.2927 | 0.5725 | 0.5825 | 0.5861 | 0.5816 |
| Pretraining & finetuning | GRU4Rec++ | 0.2665 | 0.2829 | 0.2527 | 0.2669 | 0.6467 | 0.6612 | 0.6600 | 0.6573 |
| | NARM | 0.2707 | 0.2890 | 0.2608 | 0.2733 | 0.6556 | 0.6612 | 0.6685 | 0.6629 |
| | STAMP | 0.2757 | 0.2860 | 0.2653 | 0.2753 | 0.5254 | 0.5377 | 0.5371 | 0.5346 |
| | SRGNN | 0.2853 | 0.2979 | 0.2706 | 0.2840 | 0.6263 | 0.6505 | 0.6453 | 0.6427 |
| | CORE | 0.2058 | 0.2091 | 0.1984 | 0.2040 | 0.7457 | 0.7384 | 0.7545 | 0.7466 |

Table 11: Experimental results on Task 2 phase-2 test.

| | | MRR@100 | | | | Recall@100 | | | |
|---|---|---|---|---|---|---|---|---|---|
| | Methods | ES | FR | IT | Overall | ES | FR | IT | Overall |
| Heuristic | Popularity | 0.3017 | 0.3068 | 0.2902 | 0.2989 | 0.5818 | 0.5934 | 0.5826 | 0.5863 |
| Pretraining & finetuning | GRU4Rec++ | 0.2648 | 0.2867 | 0.2569 | 0.2695 | 0.6473 | 0.6619 | 0.6600 | 0.6577 |
| | NARM | 0.2742 | 0.2938 | 0.2658 | 0.2779 | 0.6617 | 0.6624 | 0.6742 | 0.6670 |
| | STAMP | 0.2809 | 0.2922 | 0.2653 | 0.2787 | 0.5340 | 0.5400 | 0.5387 | 0.5381 |
| | SRGNN | 0.2878 | 0.3035 | 0.2701 | 0.2863 | 0.6359 | 0.6582 | 0.6491 | 0.6493 |
| | CORE | 0.2016 | 0.2138 | 0.1967 | 0.2040 | 0.7530 | 0.7387 | 0.7572 | 0.7495 |

Table 12: Full results of BLEU scores in Task 3.

| | Validation | Phase-1 Test | Phase-2 Test |
|---|---|---|---|
| mT5-small, $K = 1$ | 0.2499 | 0.2265 | 0.2245 |
| mT5-small, $K = 2$ | 0.2401 | 0.2176 | 0.2166 |
| mT5-small, $K = 3$ | 0.2366 | 0.2142 | 0.2098 |
| mT5-base, $K = 1$ | 0.2477 | 0.2251 | 0.2190 |
| Last Product Title | 0.2500 | 0.2677 | 0.2655 |

