# OpenReview forum: "Amazon-M2: A Multilingual Multi-locale Shopping Session Dataset for Recommendation and Text Generation"
_NeurIPS.cc/2023/Track/Datasets_and_Benchmarks — NeurIPS 2023 Datasets and Benchmarks Poster_

### Official Review · Reviewer_vaN1 · 2023-07-09
**Amazon-M2: A Multilingual Multi-locale Shopping Session Dataset for Recommendation and Text Generation**

**Rating:** 7
**Confidence:** 3
**Correctness:** Yes.
**Clarity:** Yes.

**Strengths:**

1. The paper presents a large session dataset with item attributes, user diversity and dataset scale
2. The dataset seems to be the first one to include the item textural description in session recommendation domain, which enables the usage of NLP technique and more aligned with the real-world scenario.
3. Comprehensive experiments indicate the new dataset provide new potential in session recommendation.


**Additional Feedback:**

No

**Documentation:**

Yes.

**Ethics:**

No.

**Limitations:**

Yes.

**Opportunities For Improvement:**


1. The fontsize in Figure 1 seems small and unclear
2. The total statisitics on #Products and #Session need replacement.
3. There is no clear discussion on the details of data collections process, i.e., whether include human annotator?

**Relation To Prior Work:**

Yes

**Summary And Contributions:**

1. The paper introduces the Amazon-M2 dataset, which addresses limitations in existing session datasets by providing comprehensive product attributes, large-scale data, multiple locales, and multilingual support. The dataset facilitates the understanding of customer shopping intentions and enables advancements in recommendation systems and text generation tasks.
2. The paper introduces a novel task called next-product title generation. Unlike traditional recommendation tasks that focus on predicting the ID or category of the next product, this task requires models to generate the actual title of the next product that has never been seen in the training set.
3.  By benchmarking representative baselines and heuristic methods, the paper empirical insights into the performance of different approaches under various evaluation metrics and novel settings.

---

> ### Author Response · Authors · 2023-08-16
> **Response to Reviewer vaN1**
>
> > **O1:**  The fontsize in Figure 1 seems small and unclear
>
> **R:**  Thanks for your suggestion. We will make font size larger in the reversion.
>
>
> > **O2:** The total statisitics on \#Products and \#Session need replacement.
>
> **R:**  Thanks for your effects on careful review, we will fix this typo in the revision.
>
>
> > **O3:** There is no clear discussion on the details of data collections process, i.e., whether include human annotator?
>
> **R:** Thanks for your suggestion on data collection process. The Amazon-M2 dataset is a collection of anonymous user session data and product data from the Amazon platform. Data is all collected by user sessions. In the data collection process, we only collect data from user behavior. We do not involve any human annotation process. All the data are collected from the user feedback.

---

> ### Comment · Reviewer_vaN1 · 2023-08-18
>
> Thank you for your reply. I have no more questions.

---

### Official Review · Reviewer_P6fP · 2023-07-20
**Review of Amazon-M2**

**Rating:** 7
**Confidence:** 4
**Correctness:** Yes.
**Clarity:** Yes.

**Strengths:**

This dataset significantly contributes to current research by filling a crucial gap, offering rich attributes on a large scale across various locales and languages, which empowers novel research directions previously unattainable. It allows for comprehensive analyses and experiments that expose the limitations of existing methods, thus highlighting areas where impactful new research can be applied. Novel tasks such as cross-locale transfer learning and text generation that are introduced have promising practical applications if resolved effectively. The dataset is particularly relevant to the research community as it could greatly benefit active research areas such as recommender systems, transfer learning, and language models. The multilingual aspect enables exploration into cross-lingual and few-shot transfer learning for recommendations, while the text attributes permit integration of the latest advances in natural language processing like large language models. In terms of research quality, the data collection process is detailed explicitly, and the analysis offers insights into the dataset characteristics. A wide range of potent baseline methods are rigorously implemented and evaluated, with the results thoughtfully analyzed to expose the limitations of current strategies. As for the ethical and social implications, the data is appropriately anonymized to protect user privacy, potential negative societal impacts are discussed, and the data diversity can contribute to fairer and less biased recommendation algorithms.

**Additional Feedback:**

1) Evaluating retrieval or ranking metrics beyond recommendation accuracy could give more insights.
2) Qualitative human evaluation of recommendations could complement quantitative metrics.


**Documentation:**

Yes.

**Ethics:**

Based on my review, I do not see any major ethical concerns with this submission that require further investigation or discussion.

**Limitations:**

The authors have made a reasonable effort to discuss the limitations and societal impact of this work transparently.

**Opportunities For Improvement:**

From a research quality standpoint, further insights could be gleaned through more rigorous hyperparameter tuning and ablation studies, while evaluating additional cutting-edge models as baselines could potentially offer a deeper understanding. Ethically and socially, there's a risk that the data could inadvertently perpetuate biases if certain demographics are overrepresented or underrepresented across locales. Finally, the impact of this work on recommender system objectives such as diversity and novelty should be analyzed.

**Relation To Prior Work:**

Yes.

**Summary And Contributions:**

This submission presents Amazon-M2, an innovative, multilingual, multi-locale shopping session dataset designed specifically for product recommendation and text generation tasks. As a key contribution, it offers an extensive, real-world dataset of customer shopping sessions from six different locales, each with unique languages, which opens up opportunities for studying diverse user-focused recommendation algorithms. Moreover, the dataset is enriched with semantic attributes such as product titles and descriptions, facilitating the utilization of advanced models that leverage text data. Through this data, three novel tasks are introduced: next product recommendation, recommendation with domain shift, and next product title generation. To demonstrate the utility of the dataset, comprehensive experiments are conducted on these three tasks, using various baselines.

---

> ### Author Response · Authors · 2023-08-16
> **Response to Reviewer P6fP**
>
> > **O1:** From a research quality standpoint, further insights could be gleaned through more rigorous hyperparameter tuning and ablation studies, while evaluating additional cutting-edge models as baselines could potentially offer a deeper understanding.
>
> **R:** Thank you for the valuable suggestion. For more rigorous hyperparameter tuning, we have added our dataset on the RecBole library (https://recbole.io/index.html), one of the most popular libraries for recommendation with 2.8k+ Github stars.  For more advanced baseline, we have added additional baseline MGS [1] in task 1 and task 2 in the revision, showing as below.
>
> **MGS results on Task 1 and task 2**
>
> | Metric            | Methods                  | UK     | DE     | JP     | Overall     |
> | :---------------- | ------------------------ | ------ | ------ | ------ | ----------- |
> | Task 1 MRR@100    |                          | 0.1668 | 0.1739 | 0.2376 | 0.1907      |
> | Task1 Recall@100  |                          | 0.5641 | 0.5479 | 0.4677 | 0.5194      |
> |                   |                          | **ES** | **FR** | **IT** | **Overall** |
> | Task 2 MRR@100    | Supervised training      | 0.2491 | 0.2775 | 0.2411 | 0.2560      |
> | Task 2 Recall@100 | Supervised training      | 0.5829 | 0.5811 | 0.5689 | 0.5766      |
> | Task 2 MRR@100    | Pretraining & Finetuning | 0.2612 | 0.2870 | 0.2693 | 0.2722      |
> | Task 2 Recall@100 | Pretraining & Finetuning | 0.5747 | 0.6133 | 0.5812 | 0.5913      |
>
>
>
> > **O2:**  Ethically and socially, there's a risk that the data could inadvertently perpetuate biases if certain demographics are overrepresented or underrepresented across locales. Finally, the impact of this work on recommender system objectives such as diversity and novelty should be analyzed.
>
> **R:** Thank you for pointing out the issue of potential bias in the dataset. First of all, we would like to highlight that Amazon-M2 reflects real-world scenarios as it collected data from diverse sources, i.e., six different locales with multiple languages with diverse user behaviors and preferences. In contrast,  prior datasets have predominantly been collected from a singular locale or country, using only one language, which results in a significant dearth of demographic diversity. Thus, Amazon-M2 may have reduced data bias due to its diverse sources. On the other hand, we also agree that there could still exist bias if certain demographics are overrepresented or underrepresented across locales. We deem it essential to incorporate fairness metrics [2,3] in evaluating algorithmic fairness across various locales. Consequently, our dataset stands to enhance the development of more fair recommendations, particularly for underrepresented locales.
>
> In addition, in Appendix D (line 740), we provide a discussion about the limitations of the proposed dataset.  We add the contents as follows in the revision:
> We also carefully consider the broader impact from various perspectives such as fairness, security, and harm to people. There may be one potential fairness issue, Amazon-M2 may inadvertently reinforce biases if specific demographics are disproportionately represented or underrepresented in different regions. While there might be fairness concerns, we assert that the Amazon-M2 dataset underscores the need for progress in fair recommendation development and serves as an effective platform for assessing fair recommendation approaches.
>
>
>
> [1] Lai et al., “An Attribute-Driven Mirror Graph Network for Session-based Recommendation,” SIGIR 2022.
>
> [2] Are we evaluating rigorously? benchmarking recommendation for reproducible evaluation and fair comparison. RecSys 2020.
>
> [3] Fairness in Recommendation: Foundations, Methods and Applications. TIST 2023.

---

> > ### Comment · Reviewer_P6fP · 2023-08-30
> >
> > Thank you to the authors for their additional efforts and clarifications.

---

### Official Review · Reviewer_7DAG · 2023-07-23
**changed from Borderline Accept to accept**

**Rating:** 7
**Confidence:** 4
**Correctness:** Overall correct.
**Clarity:** This paper has smooth language and st…

**Strengths:**

1. This paper introduces a multilingual and multi-local session dataset for recommendation and related tasks.
2. Comprehensive analysis of dataset has been provided from five different perspectives.
3. Evaluation tasks conducted on Amazon-M2 span across a wide range of fields, including recommender systems, transfer learning and natural language processing.

**Additional Feedback:**

N/A

**Documentation:**

This paper includes an introduction of the proposed Amazon-M2 dataset with sufficient details.

**Limitations:**

Same as “Opportunities For Improvement”.

**Opportunities For Improvement:**

1. The authors should consider adding more state-of-the-art session-based recommendation methods, such as MGS [Lai2022] and G3SR [Deng2022].
2. Metrics such as NDCG should be included for quantitative evaluation in recommendation tasks.
3. While Task 3, as a combination of the field of recommendation and natural language processing, is interesting, it would be beneficial to include more examples to show the comparison between generated results and expected titles.

[Lai2022] S. Lai et al., “An Attribute-Driven Mirror Graph Network for Session-based Recommendation,” Proceedings of the International ACM SIGIR Conference on Research and Development in Information Retrieval, 2022, pp. 1674–1683.
[Deng2022] Z.-H. Deng et al., “G^3SR: Global Graph Guided Session-Based Recommendation,” IEEE Transactions on Neural Networks and Learning Systems, pp. 1–14, 2022.

**Relation To Prior Work:**

This paper has provided a discussion and comparison of related session datasets for recommendation.

**Summary And Contributions:**

This paper presents a large-scale multilingual session dataset with detailed item attributes and extensive user data collected from six different locales. Based on this dataset, three subsequent tasks are designed to evaluate the potential performance of recommendation and text generation.

---

> ### Author Response · Authors · 2023-08-16
> **Response to Reviewer 7DAG (1/2)**
>
> > **O1:** The authors should consider adding more state-of-the-art session-based recommendation methods, such as MGS [Lai2022] and G3SR [Deng2022].
>
> **R:** Thank you for the great suggestions. We have added discussions on these two papers properly in Section 1 and 4. The experimental result on MGS is shown below and paper revision in Table 3 and 4. We can see that the MGS performance is comparable to other session algorithms, but no significant improvement. For the G3SR model, we have not yet obtained the open-source code, as the authors have not made it available. We are in active communication with the authors to acquire the code. Once we have it, we will promptly update our work with the experimental results in the forthcoming revision.
>
> **MGS results on Task 1 and task 2**
>
> | Metric            | Methods                  | UK     | DE     | JP     | Overall     |
> | :---------------- | ------------------------ | ------ | ------ | ------ | ----------- |
> | Task 1 MRR@100    |                          | 0.1668 | 0.1739 | 0.2376 | 0.1907      |
> | Task1 Recall@100  |                          | 0.5641 | 0.5479 | 0.4677 | 0.5194      |
> |                   |                          | **ES** | **FR** | **IT** | **Overall** |
> | Task 2 MRR@100    | Supervised training      | 0.2491 | 0.2775 | 0.2411 | 0.2560      |
> | Task 2 Recall@100 | Supervised training      | 0.5829 | 0.5811 | 0.5689 | 0.5766      |
> | Task 2 MRR@100    | Pretraining & Finetuning | 0.2612 | 0.2870 | 0.2693 | 0.2722      |
> | Task 2 Recall@100 | Pretraining & Finetuning | 0.5747 | 0.6133 | 0.5812 | 0.5913      |
>
>
>
>
> > **O2:** Metrics such as NDCG should be included for quantitative evaluation in recommendation tasks.
>
> **R:** Thank you for the great suggestions on evaluation. The Normalized Discounted Cumulative Gain (NDCG) metric is a widely used metric in information retrieval evaluation. It assesses the quality of a ranked list by considering both the relevance and the position of each item, providing a normalized measure of how well the ranked list captures relevant results. We have now added the NDCG metric in our revision for a more comprehensive comparison in Appendix C. The results are also shown as follows. The observations are similar with Recall@100.
>
> **NDCG results on Task 1**
>
> |            | UK     | DE     | JP     | Overall |
> | ---------- | ------ | ------ | ------ | ------- |
> | Popularity | 0.2065 | 0.1993 | 0.2515 | 0.2175  |
> | GRU4Rec++  | 0.2255 | 0.2184 | 0.2733 | 0.2374  |
> | NARM       | 0.2336 | 0.2248 | 0.2823 | 0.2452  |
> | STAMP      | 0.2262 | 0.2198 | 0.2726 | 0.2379  |
> | SRGNN      | 0.2325 | 0.2345 | 0.2715 | 0.2448  |
> | CORE       | 0.2698 | 0.2667 | 0.3204 | 0.2839  |
> | MGS        | 0.2314 | 0.2357 | 0.2782 | 0.2471  |
>
>
>
> **NDCG results on Task 2**
>
> |                          | Methods    | ES     | FR     | IT     | Overall |
> | :----------------------- | ---------- | ------ | ------ | ------ | ------- |
> | Heuristic                | Popularity | 0.3370 | 0.3453 | 0.3210 | 0.3339  |
> | Pretraining & finetuning | NARM       | 0.3434 | 0.3427 | 0.3620 | 0.3265  |
> |                          | STAMP      | 0.3142 | 0.3328 | 0.3015 | 0.3160  |
> |                          | SRGNN      | 0.3381 | 0.3626 | 0.3249 | 0.3418  |
> |                          | CORE       | 0.3127 | 0.3325 | 0.3085 | 0.3181  |
> |                          | MGS        | 0.3350 | 0.3491 | 0.3143 | 0.3317  |
> | Supervised Training      | NARM       | 0.3275 | 0.3481 | 0.3173 | 0.3310  |
> |                          | STAMP      | 0.3104 | 0.3306 | 0.2991 | 0.3133  |
> |                          | SRGNN      | 0.3206 | 0.3369 | 0.3083 | 0.3217  |
> |                          | CORE       | 0.3030 | 0.3272 | 0.2995 | 0.3102  |
> |                          | MGS        | 0.3189 | 0.3352 | 0.3217 | 0.3259  |

---

> > ### Author Response · Authors · 2023-08-16
> > **Response to Reviewer 7DAG (2/2)**
> >
> > > **O3:** While Task 3, as a combination of the field of recommendation and natural language processing, is interesting, it would be beneficial to include more examples to show the comparison between generated results and expected titles.
> >
> > **R:** Thanks for your great suggestion on adding a case study for task 3. We provide the following example to illustrate the quality of the generated results intuitively. The examples are also added in Figure 3 of the revision paper. We note that the generated titles look generally good, nonetheless,  the generated ones still lack details, especially numbers,e.g., (180x200cm) in example 2.  We also show the case study below.
> >
> >
> >
> > **Comparison between ground truth title and product titles generated by mT5-small, K=1.**
> >
> > | T5                                                           | **Ground Truth**                                             |
> > | ------------------------------------------------------------ | ------------------------------------------------------------ |
> > | MuyDoux Funda para Xiaomi Mi Pad 5 / 5 Pro 11 Pulgadas 2021, Tapa Frontal Lisa y Reverso Suave, Encendido/Apagado automático, Funda Ligera y Delgada de Tres Pliegues. Color: Negro | MuyDoux Funda para Xiaomi Mi Pad 5 / 5 Pro 11 Pulgadas 2021, Tapa Frontal Lisa y Cubierta Trasera Suave, Auto Sueño/Estela, Carcasa Ligera y Delgada para Mi Xiaomi Pad 5 / 5 Pro 5G, Oro Rosa |
> > | Amazon Brand - Umi Colchón de Microfibra,Cubrecolchón,Antialérgico,Suave-(135x190cm) | Amazon Brand - Umi Colchón de Microfibra,Cubrecolchón,Antialérgico,Suave-(180x200cm) |
> > | Raid Spray Insecticida - Aerosol para Moscas y Mosquitos, Eucalipto. Eficacia Inmediata, Incoloro, 400 ml | Raid ® Spray Insecticida - Aerosol para moscas y mosquitos, Frescor Natural. Eficacia inmediata. Pack de 3 Unidades, 600ml |
> > | Rapesco 1498 Carpeta Sobre Portafolios Plástica con Broche de Presión, Tamaño A5, Colores Surtidos, Paquete de 25 | Rapesco 1494 Carpeta Sobre Portafolios A4+ horizontal, Colores Surtidos Translucídos, 20 unidades |

---

> > > ### Comment · Reviewer_7DAG · 2023-08-29
> > >
> > > Thanks for your good efforts. I think it is a good work and have no further questions, raised my score to 7.

---

> ### Author Response · Authors · 2023-08-23
> **A gentle reminder for reviewer 7DAG**
>
> Dear Reviewer 7DAG,
>
> Thank you for your valuable comments on our work. We've carefully considered your feedback and have provided a comprehensive response. If you have any more questions or comments, please feel free to let us know.  We are happy to provide any further explanations if needed. Thank you once again for your great efforts.

---

### Official Review · Reviewer_hA57 · 2023-07-24
**A good dataset paper**

**Rating:** 7
**Confidence:** 4
**Correctness:** Yes
**Clarity:** Yes

**Strengths:**

1. The dataset mitigates the limitations of existing session datasets in terms of item attributes, user diversity, and dataset scale, which is innovative and important in the session-based recommendation domain.
2. In the analysis section, the authors provides comprehensive and convinced empirical insights, including an analysis of the long-tail phenomenon, product overlap between locales, session lengths, repeat patterns, and collaborative filtering patterns.
3. The paper contributes to the field by introducing three tasks based on the dataset, especially a new proposed task next-product title generation, providing a benchmark for various algorithms and drawing new insights for further research and practice.


**Additional Feedback:**

No

**Documentation:**

Yes

**Limitations:**

See the above comments

**Opportunities For Improvement:**

1. The implementation details of the proposed algorithms are not provided, especially the setup of hyperparameters, which makes it difficult to reproduce the study.
2. The way to construct distribution shift in task 2 is not too clear. It is better for the authors to clarify it in the paper.


**Relation To Prior Work:**

Yes

**Summary And Contributions:**

This paper proposes the Amazon-M2 dataset for session-based recommendation, which is a multilingual multi-locale shopping session dataset. In specific, this dataset contains millions of user sessions from six different locales in Amazon platform, particularly English, German, Japanese, French, Italian, and Spanish. Empirically, this paper introduces two typical tasks and one innovative task based on this dataset: next-product recommendation, next-product recommendation with domain shifts, and next-product title generation. Last, this paper also implements several recommendation models to complete these tasks.

---

> ### Author Response · Authors · 2023-08-16
> **Response to reviewer hA57**
>
> > **O1:** The implementation details of the proposed algorithms are not provided, especially the setup of hyperparameters, which makes it difficult to reproduce the study. T
>
> **R:** Thank you for providing these valuable suggestions. We had included the implementation details of algorithms and hyperprameter search space in Appendix A.
>
> * For task 1 and task 2, we adopted the Recbole library (https://recbole.io/index.html) [1], which is one of the most popular libraries for recommendation with 2.8K+ Github stars, to implement the baseline algorithms.  We followed its suggested library to the hyperparameter search space to tune the algorithms, as shown in Appendix A.
> * For task 3, we adopted the code example provided by HuggingFace[2] which supports text-to-text generation. The hyperparameter search space is shown in Appendix A.
>
> To ensure reproductivity, we will also open source the code repository as soon as possible, while the code repository is currently under internal review in Amazon.
>
> Moreover, to ease the better utilization of datasets, **we have added our dataset on the RecBole recommendation library.** Researchers in the recommendation domain can then easily access the dataset and investigate more baseline algorithms with standard implementation. Our proposed Amazon-M2 dataset will appear in the recent new version.
>
> [1] Zhao et al. Recbole: Towards a unified, comprehensive and efficient framework for recommendation algorithms. CIKM 2021.
> [2]https://github.com/huggingface/transformers/blob/main/examples/pytorch/summarization/run_summarization_no_trainer.py
>
>
>
> > **O2:** The way to construct distribution shift in task 2 is not too clear. It is better for the authors to clarify it in the paper.
>
> **R:** Thank you for providing valuable opportunities to improve our paper revolving on how to construct distribution shift in task 2. We  elaborate more details as follows and add those details in the Section 3 of revision (line 171-177).
>
> The distribution shifts in Amazon-M2 dataset are two-fold revolving on (1) user behavior and (2) the multilingual nature. In the following, we elaborate these two aspects.
>
> (1) The distribution shift in user behavior can be indicated by people in different locales preferring different products. Experimental evidence can be found in Figure 2(b) which shows the product overlapping ratio between different locales. The distribution overlapping ratio is calculated as $\frac{|\mathcal{N}_a \cap \mathcal{N}_b|}{|\mathcal{N}_a|}$, where $\mathcal{N}_a$ and $\mathcal{N}_b$ correspond to the products set of locale $a$ and $b$. A small overlapping ratio indicates that users in two regions prefer different products with different user behavior. x and y axes in Figure 2(b) stand for locale a and b. We can see that most overlapping ratios between the popular locales and underrepresented locales are around 0.2 indicating that there are about 80\% of products that never appear in popular locales.
>
> (2) The distribution shift can also be found due to the multilingual nature of the dataset. The product textual description from different locales are in different languages, leading to feature distribution shift. Such multilingual distribution shift is a long-established topic in NLP domain. For example,
>
> * Nzeyimana and Rubungo [3] figure out the importance of modeling morphology disparity on multiple languages to alleviate multilingual distribution shift. KinyaBERT: a Morphology-aware Kinyarwanda Language Model.
> * Rust et al. [4] identify the importance of tokenization and shows how tokenization differences in multiple languages lead to the performance disparity.  How Good is Your Tokenizer?.
>   On the Monolingual Performance of Multilingual Language Models
> * Lin et al. [5] demonstrate that both positive transfer and negative transfer exist when choosing different languages for cross-lingual transfer.
>
> [3] Antoine Nzeyimana and Andre Niyongabo Rubungo. KinyaBERT: a Morphology-aware Kinyarwanda Language Model. ACL 2022.
>
> [4] Rust et al. "How Good is Your Tokenizer? On the Monolingual Performance of Multilingual Language Models." ACL 2021.
>
> [5] Lin et al. "Choosing Transfer Languages for Cross-Lingual Learning." ACL 2019.

---

### Decision · Program_Chairs · 2023-09-22

**Decision:**

Accept (Poster)

**Comment:**

This paper proposes the Amazon-M2 dataset for session-based recommendation. All reviewers agree that the dataset addresses limitations in existing datasets by providing comprehensive product attributes, large-scale data, multiple locales, and multilingual support. Comprehensive and convincing analysis of the dataset has been provided. A wide range of evaluation tasks, including recommender systems, transfer learning and natural language processing, were also conducted. The discussion addressed the concerns during the initial review.